# The Property Characterization of α-Sialon/Ni Composites Synthesized by Spark Plasma Sintering

**DOI:** 10.3390/nano9121682

**Published:** 2019-11-25

**Authors:** Adedayo Sheriff Adeniyi, Bilal Anjum Ahmed, Abbas Saeed Hakeem, Faheemuddin Patel, Akolade Idris Bakare, Anwar Ul-Hamid, Amir Azam Khan, Muhammad Ali Ehsan, Tahir Irfan Khan

**Affiliations:** 1Mechanical Engineering Department, King Fahd University of Petroleum & Minerals, Dhahran 31261, Saudi Arabia; sherifadeniyi@gmail.com (A.S.A); faheemmp@kfupm.edu.sa (F.P.); 2Department of Mechanical Engineering, National University of Sciences and Technology, Islamabad 44000, Pakistan; banjumahmed@gmail.com; 3Center of Excellence in Nanotechnology, King Fahd University of Petroleum & Minerals, Dhahran 31261, Saudi Arabia; aibakare@kfupm.edu.sa (A.I.B.); meali@kfupm.edu.sa (M.A.E.); 4Center of Engineering Research, Research Institute King Fahd University of Petroleum and Minerals, Dhahran 31261, Saudi Arabia; anwar@kfupm.edu.sa; 5Department of Mechanical and Manufacturing Engineering, Faculty of Engineering, University Malaysia Sarawak, Sarawak 97000, Malaysia; akamir@unimas.my; 6Department of Mechanical Engineering, Faculty of Engineering & Informatics, University of Bradford, Bradford BD7 1DP, UK; t.khan20@bradford.ac.uk

**Keywords:** α-sialon, sialon–nickel composite, spark plasma sintering, microstructure, densification, mechanical properties, thermal properties

## Abstract

This study investigates the effect of micron-sized nickel particle additions on the microstructural, thermal, and mechanical property changes of α-sialon ceramic composites. The α-sialon/Ni composites were synthesized with an increasing amount of Ni (10–40 wt.%) using the spark plasma sintering technique and nanosized alpha precursors at a relatively low synthesis temperature of 1500 °C with a holding time of 30 min in each case. The density of the samples increased with the increase in Ni content of up to 15 wt.% and, with the further increase in Ni content, it became almost constant with a slight decrease. Furthermore, thermal conductivity and thermal expansion properties of Ni-sialon composites improved slightly with the inclusion of 10 wt.% Ni. The addition of Ni to α-sialon matrix resulted in a decrease in the hardness of the composites from HV_10_ 21.6 to H_V10_ 16.3, however the presence of Ni as a softer interfacial phase resulted in a substantial increase in the fracture toughness of these composites. Fracture toughness was found to increase by approximately 91% at 40 wt.% Ni addition.

## 1. Introduction

Engineering applications (such as turbine components, ball bearings, and hard cutting tools) require the use of materials that exhibit high strength, toughness and hardness, exceptional thermal performance, high resistance to oxidation, wear, and corrosion. Silicon nitride (Si_3_N_4_) is a ceramic material that has been extensively used in areas requiring exceptional mechanical properties at elevated temperature, such as gas turbine engine parts [1]. Difficulty in sintering silicon nitride material led to the development of sialon materials, where Si_3_N_4_ structure is modified by replacing a fraction of the silicon (Si) and nitrogen (N) with aluminum (Al) and oxygen (O) [1,2]. Sialon is a solid solution of Si_3_N_4_ and Al_2_O_3_ [1,3,4]. Sialons could primarily exist in two major phases: alpha or beta [1,3,5,6,7,8]. The β-sialon, developed during the 1960s, is represented with the general formula of Si_6-z_Al_z_O_z_N_8-z_, where 0 < z < 4.2. The α-sialon, which was developed later on, is represented by Me_m/v_Si_12-(m+n)_Al_m+n_O_n_N_16-n_, where ‘Me’ could be a rare-earth metal (Ln), an alkaline earth metal (Mg, Ca, etc.), or a metal belonging to the alkali group (Li). The ‘m’ and ‘n’ values represent the (Al–N) and (Al–O) bonds, which substitute the (m+n): (Si-N) bonds, and ‘v’ represents the valency of cation ‘Me’ [5]. The performance of α- and β-sialons have been studied extensively, and they have been discussed mainly based on their microstructure–property relationship. The elongated grains in β-sialon are responsible for their improved fracture toughness and, similarly, the improved hardness of the α-sialon is attributed to their equiaxed morphology [9]. It is desirable to obtain α-sialons with a high fracture toughness, and this has been achieved in several studies [10,11]. Various reinforcements and additives, such as cubic boron nitride (cBN) [12,13], Ca metal [14,15,16], Al metal [17], AlN [18], SiC [15] and rare-earth oxides [19,20,21], have been used to reinforce α-sialon composites with improved fracture toughness. However, to the best of our knowledge, little or no work has been reported on the influence of Ni inclusions on the behavior of α-sialons. 

Spark plasma sintering (SPS) [17,22,23,24], commonly known as field assisted sintering technique (FAST), is used to synthesize a number of difficult-to-sinter materials [25]. With the ability to sinter at high heating rates, this technique has been used to develop bulk ceramic materials in a short time. Al_2_O_3_ [26], ZrO_2_ [27], as well as sialon- [17] based ceramics, have been sintered using SPS technique. This technique, when compared to conventional sintering techniques, is used to develop ceramics with high densification and better thermal and mechanical properties. 

Improvements in thermal, mechanical, and wear properties have been reported in the literature due to the incorporation of nickel particles in alumina [28,29,30]. However, improvements in mechanical properties by adding Ni in sialon matrix have not been studied so far. In the present work, our goal was to investigate the effect of increasing Ni metal content on the microstructure, densification, mechanical, and thermal properties of α-sialon ceramic in order to evaluate their viability for industrial applications, such as gas turbine engine parts.

## 2. Materials and Methods

A mixture of starting materials was reacted together for preparation of α-sialon composite samples. The α-sialon matrix consisted of commercially available powders (Table 1), namely CaO (<160 nm, 98%, Sigma Aldrich, St. Louis, MO, USA), SiO_2_ (20–50 nm, 99.5%, Sigma Aldrich, St. Louis, MO, USA), AlN (<100 nm, Sigma Aldrich, St. Louis, MO, USA), and Si_3_N_4_ (300–500 nm, 95% α-phase content, UBE Industries SN-10, Tokyo, Japan). A varying amount of micro-sized Ni metal powder (<100 µm, 99.99%, Fisher Scientific, Pittsburgh, PA, USA) was used as an additive in the composite. Table 1 summarizes the weight percentages (wt.%) of the powder precursors in each composition. A composition was based upon the general formula, Ca_m/2_ Si_12-(m+n)_ Al_(m+n)_O_(n)_ N_16-n_, where m = 1.6 and n = 1.2. Probe sonicator (Model VC 750, Sonics, Newtown, CT, USA) was employed to achieve uniform mixing of the powders in ethanol. After 20 min of probe sonication, the powder mixture was oven-dried at 90 °C temperature for about 20 h to evaporate the ethanol. After drying, the mixtures were consolidated using spark plasma sintering (SPS) equipment (FCT system, model HP D5, Frankenblick, Germany). The powders were synthesized in a 20 mm graphite die under a uniaxial pressure of 50 MPa. The synthesis temperature, heating rate, and soaking time at which all the samples were sintered were 1500 °C, 100 °C/min, and 30 min, respectively [28,29]. All experiments were carried out in a vacuum of 5 × 10^−2^ mbar. An SPS curve showing different stages of the sintering process is shown in Figure 1. The insert represents the displacement vs. temperature/time cycle of sample S2 (Table 1).

Subsequent to the SPS processing, the graphite contamination at the surface of the sintered samples was removed using SiC abrasives (grit sizes ranging from 180 grit to 1200 grit). Furthermore, the samples were ground to a finer finish using a diamond grinding wheel. To obtain a mirror-like surface, the ground samples were polished using alumina suspension (particle size 0.3 µm). 

The microstructure was observed using a backscattered mode of field emission scanning electron microscope (FESEM Lyra3, Tescan, Brno, Czech Republic) equipped with energy-dispersive X-ray spectroscopy (EDXS, silicon drift detector X-Max^N^, Oxford Instruments, High Wycombe, UK). Skyscan1172 Micro-CT (Bruker, Kontich, Belgium) equipment was used to analyze the distribution of the Ni metal particles in the sialon matrix. Phase analysis was performed using X-ray diffraction (XRD) equipment (Rigaku MiniFlex X-ray diffractometer, Tokyo, Japan). The diffractometer was operated at a wavelength, current, and voltage of 0.15416 nm, 10 mA, and 30 kV, respectively. High-temperature XRD analysis was performed to ascertain the evolution of phase(s). The established Archimedean principle of density measurement was employed to determine the densities of the sintered composites. A Mettler Toledo kit was used to determine the density using distilled water as the immersion medium. The average of five results was presented. A universal hardness tester (developed by Zwick-Roell, ZHU250, Ulm, Germany) was used to determine the hardness of the sintered samples at a load of 10 kg. Using the maximum crack length (*d*) and the hardness value obtained, the indentation fracture toughness (*K_1C_*) was obtained using the Evans criterion, where ‘*MCL*’ represents the maximum crack length and ‘*d*’ represents the average diagonal length [27]. 

(1)KIC=0.48(MCLd2)−1.5 HV10d23

Thermal expansion equipment (Mettler Toledo, TMA/SDTA-LF/1100, Greifensee, Switzerland) was used to compute the coefficient of thermal expansion of the synthesis samples. The samples used for this experiment were cut to finished dimensions of 4 mm × 4 mm × 4 mm each. Thermal conductivity (c-TERM TCi, Fredericton, NB, Canada) of the specimens was measured at room temperature by applying a transient but constant heat to the sintered ceramic specimens via a one-sided interfacial heat reflectance sensor. 

## 3. Results and Discussion

### 3.1. Microstructure and Phase Analysis

FESEM micrographs depicting the fracture surface morphology of the ceramic composites are shown in Figure 2a–g. Figure 2a presents the microstructure of pure α-sialon (Sample ID S1) that was not incorporated with Ni metal particles. The microstructure consisted of distinct single phase-α-sialon grains (equiaxed shape), as observed in Figure 2a. A micrograph of sample S1 was compared with other samples containing Ni inclusions, wherein little or no porosity was observed. Figure 2b–g represent the samples S2–7 with a microstructure consisting of bright (white) Ni particles, uniformly distributed, and embedded within a gray sialon matrix. However, some of the Ni inclusions were seen to tear off from the α-sialon matrix and left an impression of a pore, as indicated by red circles in Figure 2b,c. The α-sialon grains in S1 depicted hexagonal/equiaxed morphology, and the addition of the Ni particles did not affect the morphology of the alpha grains. The presence of Ni site vacancies was observed to be the least for sample S7, indicating its high fracture toughness. This was attributed to the Ni particles embedding in the ceramic matrix and filling voids between the grains of the α-sialon composite. Micro-CT images of the α-sialon samples containing 10 and 40 wt.% Ni are shown in Figure 3a,b, respectively. The nickel particles can be seen distributed throughout the sialon matrix and this distribution was more uniform at lower Ni content.

Figure 4 presents the room-temperature X-ray diffractograms of all composites sintered at 1500 °C. Single-phase α-sialon peaks can be observed for S1. The increasing content of Ni particles did not cause any phase transformation in the α-phase of sialon. However, it has been reported in the literature that, as a result of reinforcing particles, α to β phase transformation can take place in α-sialon materials when sintered between 1350 and 1600 °C [30,31,32]. The XRD plot for the α-sialon/0–40 wt.% Ni confirms the presence of the Ni phase and Ni did not react with any other reactant from the powder mixture to form any compound (such as Ni_3_Al or Ni_2_Si) as per XRD analyses. The synthesized composite samples were thermally stable as no transformation was observed with increasing Ni content [33].

In situ XRD analysis was performed at different temperatures. Figure 5 shows the high-temperature XRD plots of α-sialon 40 wt.% Ni composites recorded at various temperatures up to 1500 °C. A mixture of the precursors was heated (continually) at different temperatures in high-temperature XRD under argon atmosphere. The crystalline phases observed are presented as a function of the temperature in Figure 5a. It was difficult to label all the constituents of the mixture as Si_3_N_4_ was a dominating powder in the mixture and a major peak of Ni was visible (2θ = 44.5°) at 100 °C. The XRD analysis in Figure 5a shows that the change in the powder mixture started at a temperature of about 1200 °C. This corresponds to the formation of the liquid phase, which was observed to take place between 1200 to 1300 °C as confirmed by the shrinkage curve in Figure 1. At 1400 °C (Figure 5a, curve No. 6), the solution/precipitation mechanism took place [34]. This corresponds to the phenomenon where raw materials, Si_3_N_4_ as well as AlN, start to react with the rest of the liquid. Other unreacted precursors, particularly AlN and intermediate phase(s), disappeared in the range 1400 to 1500 °C. The amount of α-sialon increased with increasing temperature until near-complete conversion at 1500 °C according to Figure 5a (7-SPS 1500 °C) obtained from Figure 4, in which a sample was spark plasma sintered at 1500 °C and XRD were recorded at room temperature. In light of the previous studies and present work, the following phase change reactions are expected to have occurred [17,22,23]:Si_3_N_4_(SiO_2_) + AlN(Al_2_O_3_) + SiO_2_ + CaO + Ni → Ca-sialon(liq.) + Ni (1300 °C)(2)
Si_3_N_4_ + AlN + CaSialon(liq.) + Ni → α-sialon + Ca-sialon(liq.) + Ni(liq.) (1400 °C)(3)
Alpha sialon + CaSialon(liq.) + Ni(liq.) → α-sialon + Ni(liq.) (1500 °C),(4)

Both (SiO_2_) and (Al_2_O_3_) represent the oxide layers present on the Si_3_N_4_ and AlN (< 2%) surfaces, respectively.

Reaction (2) occurs at ~1300 °C, Reaction (3) at about 1400 °C, and Reaction (4) at 1500 °C. Both SiO_2_ and Al_2_O_3_ in Equation (2) are present on the surface of the starting nitride powders and in the presence of CaO from a liquid phase (generally called the intergranular glassy phase), and subsequently aid to form a solid solution of silicon nitride (sialon). As viscous liquid phase increases, grain boundary diffusivity and a solution–diffusion–reprecipitation process occurs. It is suggested that the reaction sequence starts with oxide formation, in which, first, a liquid phase is formed, resulting in the formation of sialons with calcium ions in the structure (Figure 1). However, the reaction kinetics may vary a little in the SPS synthesizing technique compared with HR-XRD.

### 3.2. Densification, Thermal and Mechanical Properties

Table 2 presents the densities of the sialon/Ni composites as a function of the weight percentage of Ni particle content. The density of α-sialon sample sintered at 1500 °C was about 3.16 g/cm^3^, which is about 99% that of the theoretical value. The density of hot-pressed Ca-sialon at 1800 °C was reported to be 3.20 g/cm^3^ [30]. When α-sialon was sintered using SPS at 1450 and 1600 °C, the reported densities were 3.11 and 3.17 g/cm^3^, respectively [17]. The measured density values for pure α-sialon in this study are consistent with reported data.

The relative density of the composites was observed to decrease with the increase in nickel content. This can be attributed to the softening of Ni particles when the composite was synthesized at 1500 °C. A possible reason for the decline in relative density is because of the difference in coefficient of thermal expansion of sialon and nickel, which probably resulted in the formation of voids during the cooling process. Moreover, nickel was observed to form a layer on the surfaces of the synthesized samples, indicating that some of the Ni droplets melted and came out from the graphite die during the SPS holding time at 1500 °C. Similar observations were reported in the literature [16,35,36,37,38]. The amount of nickel that melted increased as the nickel content was increased and, hence, resulted in comparatively lower relative densities (as the actual amount of Ni in the final composite was lower as compared to the initial value of Ni, which was used to calculate the theoretical density).

Thermal conductivity was determined at 25 °C as a function of nickel content. Thermal conductivity of 10% Ni-α-sialon composites (5.81 W/m·k) was observed to be slightly higher than the pure α-sialon sample (5.67 W/m·k). However, we did not observe any appreciable effect on thermal conductivity value by further increasing Ni content, as shown in Table 2. This was due to no actual increase in the Ni content as, when the metal amount increased, more liquid Ni drained out from the graphite die at 1500 °C. Furthermore, transport phenomena attributed two different materials across an interface between the matrix materials and inclusions. The thermal conductivity of a composite depends on the presence of voids, interfacial resistance, and the particle size of both the inclusions and matrix [24].

Thermal expansion of 10% Ni-sialon composites (3.17 ppm·K^−1^) was observed to be higher than the pure α-Sialon sample (2.62 ppm·K^−1^). This was due to nickel inclusions having a relatively high thermal expansion value of 13–15 ppm·K^−1^(Ni). With further increase in the nickel content, the thermal expansion was seen to decrease. Increasing the nickel content of the composite coincided with the appearance of nickel in the voids of the composite and apparently reduced the effective thermal strain and lowered the coefficient of thermal expansion of the composite [39].

The hardness of all the samples ranged between 16 to 21 GPa. It was observed that, with increasing Ni content in the matrix of the α-sialon, the hardness value decreased, with the lowest of about 16 GPa for the sample with 40 wt.% Ni. As expected, nickel-induced reduction in the relative density of Ni-sialon composites also reduced the hardness of the composites. A similar effect of a decrease in hardness of sialon composites was observed with increasing Fe/Mo metal content [40].

The fracture toughness of Ni-sialon composites increased with the increase in the nickel content. Fracture toughness of 7.3 MPa √m was recorded for pure sialon and, with the increase in Ni content to 40 wt.%, fracture toughness was seen to increase by approximately 91% to a value of 14.1 MPa √m. It has been reported that the primary toughening mechanism in ceramic–metal composites involves the stretching and deformation of inclusions [41]. Cracks gets deflected and blunted when reaching a ceramic–metal interface due to the difference between the abilities to deform a ductile inclusion and a brittle matrix (see Figure 6). It is well established that wetting of ceramic by metallic melt is usually weak [42]. However, the pores present in sialon matrix permit Ni to propagate through matrix grains, thereby improving the wettability with the increase of Ni content. In this study, the presence of nickel in voids between the sialon grains is thought to improve the bonding of the grains within the ceramic matrix at the interface, and this results in the improvement in fracture toughness values. Similar improvement in the fracture toughness of the composites as a result of enhanced interfacial bonding strength has also been reported earlier [43]. A similar increase in fracture toughness has also been reported for alumina reinforced Ni composites [44].

## 4. Conclusions

The effect of increasing the amount of Ni (10–40 wt.%) on microstructural, thermal, and mechanical properties of α-sialon (Ca_0.87_Si_10_Al_3.04_O_1.30_N_16.09_) ceramic composites was studied for the first time. The Ni-α-sialon composites were consolidated using nanosized alpha precursors SPS at 1500 °C for 30 min under 50 MPa. Thermal conductivity and thermal expansion of Ni-α-sialon composites were found to improve slightly with the inclusion of Ni particles, however no appreciable effect on thermal conductivities and thermal expansion were observed when Ni content increased beyond 10 wt.%. The addition of Ni to the α-sialon matrix was found to decrease the hardness of the composites. A substantial increase in the fracture toughness of Ni-α-sialon composites was observed. Fracture toughness was found to increase by approximately 91% as the Ni content increased up to 40 wt.%. The synthesized sialon/nickel composites offer improved fracture toughness, which is beneficial for applications as cutting tools, body armor ceramic plates, hip replacement prosthetics, spark plug insulators, and brakes for aircrafts.


**Highlights**


α-sialon/nickel composites were synthesized by spark plasma sintering (SPS) using nanosized alpha sialon starting powder precursors and micron-sized nickel particles.Effect on different properties of Ni-sialon composites on addition of Ni was evaluatedThermal conductivity, thermal expansion, theoretical density, and fracture toughness increased with the addition of nickelRelative density and Vickers hardness decreased with the addition of nickel

## Figures and Tables

**Figure 1 nanomaterials-09-01682-f001:**
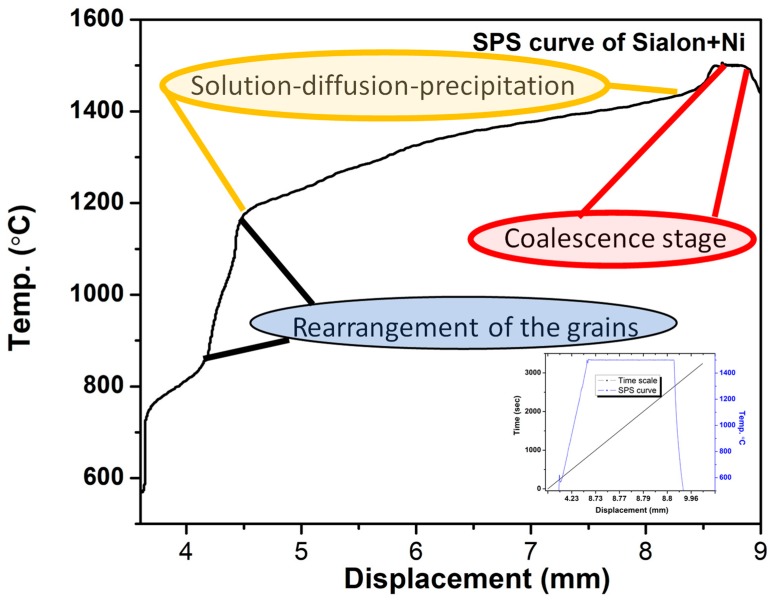
Spark plasma sintering (SPS) densification curve illustrating the synthesis mechanism of Ni + sialon composite (sample S2).

**Figure 2 nanomaterials-09-01682-f002:**
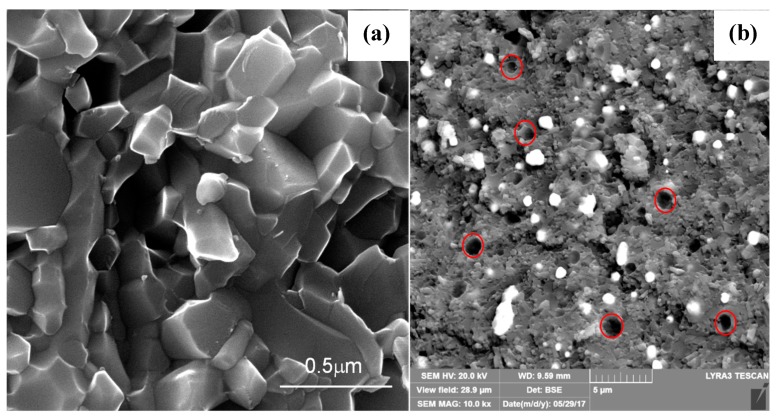
FESEM micrographs (**a**–**g**) of the sintered α-sialon samples with various wt.% Ni contents of 0, 10, 15, 20, 25, 30, and 40, respectively. The gray region reveals the sialon matrix, while the white region represents Ni particles dispersed in the sialon matrix. The red circles in figures (**b**) and (**c**) indicate the pinholes developed upon fracture due to the pullout of Ni particles.

**Figure 3 nanomaterials-09-01682-f003:**
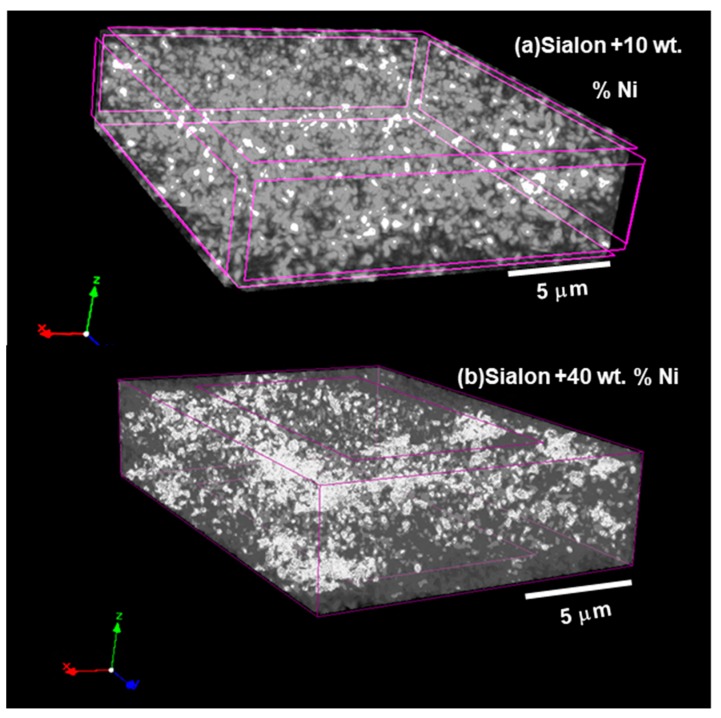
Micro-CT images of the alpha (α)-sialon samples containing (**a**) 10 and (**b**) 40 wt.% Ni.

**Figure 4 nanomaterials-09-01682-f004:**
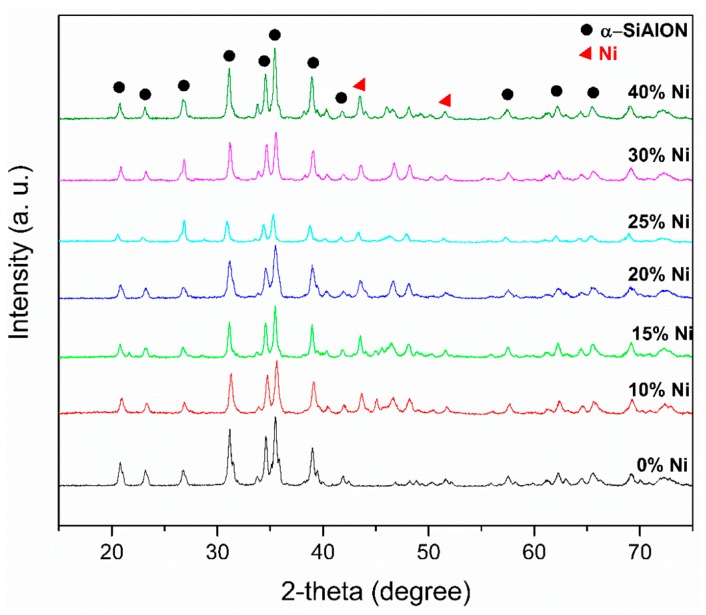
Room temperature XRD plots of the sintered Sialon samples with different Ni content.

**Figure 5 nanomaterials-09-01682-f005:**
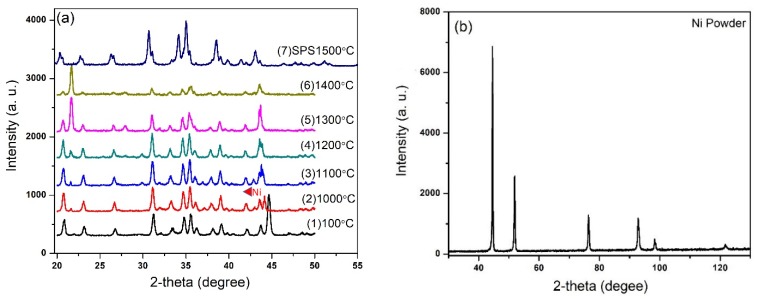
(**a**) High-temperature XRD plots of the sintered Sialon with 40 wt.% Ni and (**b**) XRD of nickel powder at room temperature.

**Figure 6 nanomaterials-09-01682-f006:**
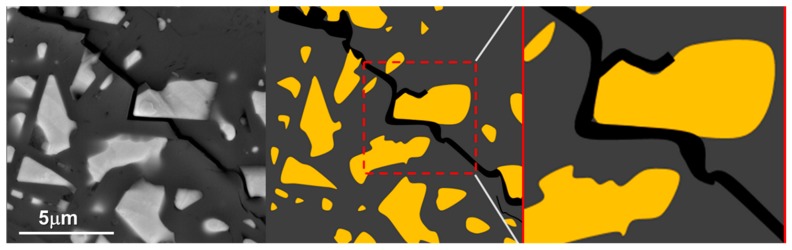
Image shows FESEM micrograph of the crack propagation path within the sialon/Ni matrix. FESEM image, along with the schematic, represents the toughness strengthening mechanisms of crack deflection and blunting.

**Table 1 nanomaterials-09-01682-t001:** The weight percentage of the Ni (micron-size) additive and α-sialon powder precursors.

Sample No.	Ni wt.%	CaO wt.%	SiO_2_ wt.%	AlN wt.%	Si_3_N_4_ wt.%
S1	0	7.572	2.028	19.370	71.030
S2	10	6.815	1.825	17.433	63.927
S3	15	6.436	1.724	16.464	60.375
S4	20	6.058	1.622	15.496	56.824
S5	25	5.679	1.521	14.527	53.272
S6	30	5.300	1.420	13.559	49.721
S7	40	4.543	1.217	11.622	42.618

**Table 2 nanomaterials-09-01682-t002:** Properties of the samples sintered at 1500 °C by SPS. Sample IDs are according to Table 1.

Sample IDs	S1	S2	S3	S4	S5	S6	S7
Ni wt.%	0	10	15	20	25	30	40
Density (g/cm^3^)	3.16(4)	3.35(6)	3.39(4)	3.33(8)	3.30(6)	3.30(7)	3.30(6)
Theoretical Density (g/cm^3^)	3.25(6)	3.47(5)	3.59(4)	3.72(5)	3.86(6)	4.01(8)	4.36(4)
Relative Density * (%)	98	97	95	90	86	83	76
Thermal Conductivity (W/m·k) **	5.67	5.81	5.78	5.74	5.80	5.81	5.81
Thermal Expansion (ppm·K^−1^)	2.62	3.17	2.96	2.89	2.84	2.75	2.70
Hardness H_V10_ (GPa)	21.6(6)	18.3(8)	17.5(5)	17.1(6)	16.6(7)	16.5(7)	16.3(5)
Fracture Toughness K_1c_ (MPa*m^1/2^)	7.3(6)	7.8(8)	8.2(4)	8.8(7)	10.2(5)	12.1(5)	14.1(6)

* Density of Ni 8.96 g/cm^3^ ** Thermal Conductivity at 25 °C.

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
