# Peer review of "The Property Characterization of α-Sialon/Ni Composites Synthesized by Spark Plasma Sintering"

_nanomaterials, 2019, doi:10.3390/nano9121682_

Round 1

Reviewer 1 Report

Authors presents the results of their works on the production of composites consisted of
α-sialon ceramic and micro-sized nickel particles. The spark plasma sintering technique have been used in synthesis of studied composites, An important issue studied by the authors was to determine the impact of Ni particle content on the microstructure, mechanical properties and thermal conductivity.

The paper requires the following amendments:

The overall remark: Authors used Ni micro-sized particles in order to enrichment of the ceramic matrix - are these nanocomposites, as they write in line 33?

In science publications, especially materials publications, the SI units should be used. Therefore such parameters as temperature, pressure, etc. must be expressed in SI units. In the presented paper authors use [oC], but in the line 236; the thermal expansion are presented in ppm·K-1 (SI unit). I propose to exchange in the whole paper, the temperature units as complied with SI system.

Introduction: the goals of research were not clearly defined.

Materials and methods: there is no information about the scanning electron microscope used and the imaging methods used for the samples tested.

Results and discussion: Figures 2; SEM images are illegible, it is not known what they depict (no descriptions on images).

Figures 3; I suggest using the arrows to explain, what the white areas does it mean. Legend - lack Ni symbol.

There are no exact results of testing the mechanical properties of the materials produced. Presenting the scope of hardness changes is not enough.

Conclusions: The lack of the general conclusion/conclusions about the importance of the results obtained and also information, where they can be used.

In my opinion, the reviewed paper requires the major alterations before being recommended for the publication in Nanomaterials.

Reviewer 2 Report

It is a good and consistent paper.

In future, I would suggest Authors

undertake numerical studies of the problem.

I think the paper can appear in the Journal as is.

Reviewer 3 Report

An interesting and pertinent paper, but it requires some minor changes before publication, in addition to a good check of the use of English by a native speaker - some examples of erroneous English statements are:

1) The first sentence of the abstract should be: "This study investigates the effect of the addition of micron-sized nickel particles on the microstructural, thermal and mechanical property changes..."

2) First sentence in the Introduction should also list "toughness" as a necessary property.

3) Introduction: "...is one of the ceramic materials that has.."

4) line 70: "...fast, assisted-sintering technique.."

5) In figure 5, the red circles need more explanation

etc...

Furthermore, please check or revise the following:

1)  The highlights and the title mention a few times "nanocomposites" but the paper mainly deals with micron-sized Ni added to SiAlON. The results also show that most grains are more than 0.3nm in size which are strictly "sub-micron" or "micron" (see e.g. 2g) in size. "Nano"  is generally understood to mean <100nm - please revise accordingly

2) In page 4 you mention that you used the Evans (or Palmqvist) method for measuring toughness but you do not mention how you ensured that all pre-existing surface stresses were removed from the surface prior to measuring the maximum crack length around the indentations. Generally this is done by a careful study of how much grinding and polishing is necessary to ensure that the cracks are not affected by pre-existing stresses. Any compressive stresses remaining in the surface will give you artificially shorter cracks which will give you erroneous increase in toughness. This is particularly worrying in the case of S6 and S7 where the densities are very low but the fracture toughness seems very high, even though the samples must contain a lot of porosity and microcracks. Did you try to corroborate these toughness values with another method? Can you see any toughness mechanisms (w.g. crack bridging, fricitional pull-outs etc) which would confirm such high toughness? In any case, the Evans/Palmqvist method is not recommended for materials with a large amount of porosity or flaws, so it should be used only for materials S1-S3.

3) In lines 144-145, you say that the "...nickel particles can be seen uniformly distributed throughout the sialon matrix..." but the photos in figures 3a and 3b indicate the opposite, i.e. the distribution is not uniform.. Can you justify this statement quantitatively?

4) In page 10 you mention that you observed no reaction between the SiAlON and the Ni particles even at 1550C. This is very surprising especially since there is a liquid phase present at lower temperatures and Ni is quite reactive. Did you check whether the Ni particles were oxidized? Please expalin these results as they are not easy to accept.

5) Please give some explanation of how do the thermal conductivity and thermal expansion results remain approximately constant even though the relative density decreases so much (i.e. the porosity increases substantially)

6) In lines 250-252 you mention that Figure 2b shows crack blunting, but this is not evident from that photo. Please give a higher magnification photo showing some micromechanisms of energy dissipation to support your results.

Round 2

Reviewer 1 Report

Dear Authors,

Thank you for considering my comments.
I accept this manuscript in present form and propose to publish it in Nanomaterials.

Reviewer 3 Report

There is still some confusion regrading the effect of Ni additions on the fracture toughness.

In your figure 6, Ni inclusions are clearly NOT adhering to the matrix, therefore they can, at most, be responsible for a small increase in toughness only indirectly, by encouraging crack wandering, as you correctly state "Cracks gets deflected and blunted when reaching a ceramic-metal interface due to the difference between the abilities to deform a ductile inclusion and a brittle matrix".

But your sentence "In this study, the presence of nickel in voids between the sialon grains is thought to improve the bonding of the grains within the ceramic matrix at the interface and this results in the improvement in fracture toughness values" is NOT supported by your results.

Looking at your results, I'm afraid I can only conclude that your perceived increased fracture toughness can only be ascribed at your inadequate preparation of your surface which did not eliminate the compressive stresses that always occur on the surface of such materials after cutting, grinding.

The only way to be sure that fracture toughness has increased is either by carefully preparing the surface till all compressive stresses have been eliminated or by by measuring it with a direct method, e.g. SENB or by looking at energy dissipation during bending tests.